# Saving 100x Storage: Prototype Replay for Reconstructing Training Sample Distribution in Class-Incremental Semantic Segmentation

**Jinpeng Chen[1], Runmin Cong[2]\*‡, Yuxuan Luo[1], Horace Ho Shing Ip[1]†, and Sam Kwong[3]‡**

[1]Department of Computer Science, City University of Hong Kong, Hong Kong
[2]School of Control Science and Engineering, Shandong University, Jinan, Shandong, China
[3]Lingnan University, Hong Kong

`jinpechen2-c@my.cityu.edu.hk, rmcong@sdu.edu.cn, yuxuanluo4-c@my.cityu.edu.hk,`
`horace.ip@cityu.edu.hk, samkwong@ln.edu.hk`

## Abstract

Existing class-incremental semantic segmentation (CISS) methods mainly tackle catastrophic forgetting and background shift, but often overlook another crucial issue. In CISS, each step focuses on different foreground classes, and the training set for a single step only includes images containing pixels of the current foreground classes, excluding images without them. This leads to an overrepresentation of these foreground classes in the single-step training set, causing the classification biased towards these classes. To address this issue, we present STAR, which preserves the main characteristics of each past class by storing a compact prototype and necessary statistical data, and aligns the class distribution of single-step training samples with the complete dataset by replaying these prototypes and repeating background pixels with appropriate frequency. Compared to the previous works that replay raw images, our method saves over 100 times the storage while achieving better performance. Moreover, STAR incorporates an old-class features maintaining (OCFM) loss, keeping old-class features unchanged while preserving sufficient plasticity for learning new classes. Furthermore, a similarity-aware discriminative (SAD) loss is employed to specifically enhance the feature diversity between similar old-new class pairs. Experiments on two public datasets, Pascal VOC 2012 and ADE20K, reveal that our model surpasses all previous state-of-the-art methods. The official code is available at `https://github.com/jinpeng0528/STAR`.

## 1 Introduction

Typically, neural networks [6, 15, 16] are trained in a one-time fashion.When data updates occur, retraining on the entire dataset is often necessary, as finetuning solely on new data leads to catastrophic forgetting [21] of previously learned knowledge. However, frequent retraining is time-consuming and requires the storage of all past data. Consequently, there is a growing interest in enabling networks to achieve continual learning, which aims to attain performance comparable to one-time training by progressively learning from new data.

---

\*Runmin Cong is also with the Key Laboratory of Machine Intelligence and System Control, Ministry of Education, Jinan, Shandong, China.
†Horace Ho Shing Ip is also with the Centre for Innovative Applications of Internet and Multimedia Technologies, City University of Hong Kong, Hong Kong.
‡Corresponding authors.

37th Conference on Neural Information Processing Systems (NeurIPS 2023).

While most continual learning research has focused on image classification, the growing expectation for computers to perform various tasks has driven the expansion of research into other domains. One such domain that has gained considerable attention is class-incremental semantic segmentation (CISS). In CISS, the training process involves multiple steps, with each step focusing on different classes. During each step, only a part of the classes is considered foreground and assigned ground-truth labels, while foreground classes from past or future steps are labeled as *background*. Compared to other continual learning tasks, CISS must address not only catastrophic forgetting but also background shift [2]. The latter refers to the fact that the background in the current step may include pixels from past or future foreground classes, causing its semantics to change across steps. Without specific countermeasures, this can greatly confuse the network and result in poor performance.

Previous CISS methods [1, 2, 3, 9, 22, 23] effectively addressed catastrophic forgetting and background shift, but they often neglect another critical issue: the discrepancies in class distributions between single-step training sets and the complete dataset. Single-step training sets include only images with pixels from corresponding foreground classes, resulting in an increased proportion of these classes compared to the complete dataset. For instance, under a CISS protocol on the Pascal VOC 2012 dataset [11], *overlapped* 19-1, 19 foreground classes are trained in the first step, and only *tv/monitor* is in the second step. In this protocol, *tv/monitor* constitutes 16.5% of pixels in the second step's training set but only 0.8% in the complete dataset. Hence, the network may become biased towards *tv/monitor*, causing numerous false positives, where pixels not belonging to *tv/monitor* are misclassified as such. Though adjusting class weights in the loss function can mitigate this issue, the limited quantity and richness of negative samples (pixels not *tv/monitor*) impede a full grasp of their potential characteristics.

To tackle this issue, we propose a method called STAR (STorage sAving Replay for CISS). After each training step, STAR stores key information, including a compact prototype for each current foreground class, which is derived from the features of this class, and statistical data related to the prototypes or class proportions. Using this information, STAR replays past foreground classes and repeats background pixels at appropriate frequencies during subsequent steps, enabling a thorough reconstruction of single-step training sample distributions with minimal storage cost. Compared to methods that save and replay raw images [1, 3], STAR significantly reduces storage requirements (using only ~1% of space) while achieving better performance. It also avoids ethical concerns, as the stored data does not contain personal information. Note that although previous replay-based CISS works [1, 3, 20] consider the main function of replay to be combating forgetting, we argue that its more crucial role is in learning new classes, as confirmed by [1, 3], where the versions with replay improve the performance on new classes more than old classes. Since new classes are typically trained individually or with few other new classes, there is a larger overrepresentation in the single-step training set, leading to a noticeable bias in the new-class classifiers that cannot access enough negative samples. The replayed samples, however, can alleviate this problem by serving as negative samples. Extending this idea, we propose to thoroughly align the class distributions of each single-step training set with the complete dataset up to that step by controlling the replay frequency.

In addition, to coordinate with our replay strategy as well as reduce forgetting, it is important to maintain old-class features during subsequent steps. Otherwise, old-class features will deviate from the saved prototypes, rendering the replay ineffective. To address this, we propose an old-class features maintaining (OCFM) loss, which is a spatially-targeted knowledge distillation that constrains the change of old-class features while allowing sufficient flexibility to learn new classes. Moreover, we introduce a similarity-aware discriminative (SAD) loss to selectively enhance discriminability between similar new and old class features, making it easier for the classifiers to distinguish them.

In summary, this paper presents the following contributions: **(1)** We propose STAR, a method that stores necessary statistical data and compact prototypes for learned foreground classes, enabling a comprehensive reconstruction of single-step training sample distributions to align with the complete dataset, thus eliminating the bias towards specific classes. **(2)** We introduce an old-class features maintaining (OCFM) loss that retains learned knowledge in a spatially targeted manner, preserving old-class features while ensuring sufficient flexibility for learning new classes. We also design a similarity-aware discriminative (SAD) loss to specifically enhance the discriminability of features between similar old-new class pairs, facilitating the classification. **(3)** On two public datasets, Pascal VOC 2012 [11] and ADE20K [34], our STAR achieves state-of-the-art performance in CISS.

## 2 Related work

### 2.1 Continual learning

Continual learning aims to enable models to incrementally learn new knowledge while avoiding catastrophic forget old knowledge. In computer vision, most continual learning research focuses on image classification and falls into three categories: regularization-based, rehearsal-based, and structure-based methods. Regularization-based methods [17, 8, 10, 4] apply constraints to parameter updates to preserve crucial learned knowledge. Rehearsal-based methods [26, 27, 24] store a small amount of past data or generate additional old-class data for replay during subsequent steps to minimize forgetting. Structure-based methods [19, 28, 18] allocate extra parameters for new classes and keep parameters related to old classes unchanged to preserve knowledge for both.

Similar to our approach, several previous continual learning studies have also aimed at enhancing memory efficiency [14, 30, 32]. For example, [14, 30] utilize compact prototypes to retain old class information and have delved into addressing prototype ineffectiveness caused by shifts in the feature space during incremental training. [32] leverages low-fidelity exemplars rather than prototypes, with a focus on narrowing the domain gap between these low-fidelity exemplars and raw images. Our method distinguishes itself in four aspects: (1) We save a single compact prototype per class for better memory efficiency, unlike some prior works [14, 32] which store multiple prototypes or exemplars. (2) We incorporates class distribution during replay, effectively rectifying the classification bias. (3) We store feature statistics beyond prototypes, offering more comprehensive cues of past classes. (4) All these studies concentrate on continual image classification, but our focus is CISS.

### 2.2 Class-incremental semantic segmentation

In recent years, a growing number of methods have focused on the CISS task. ILT [22] employs knowledge distillation on both output and intermediate features to reduce catastrophic forgetting. MiB [2] addresses background shift for the first time, using operations on predicted probabilities to eliminate semantic contradictions between steps. SDR [23] leverages prototype matching and contrastive learning to enhance discrimination between features of different classes while applying feature sparsity techniques to reserve space for future knowledge. PLOP [9] preserves old-class knowledge using pseudo labels generated by the previous model for supervision and a spatially-aware variant of POD [10]. SSUL [3] introduces an extra unknown class aided by salient object detection models to tackle background shift. DKD [1] adopts decomposed knowledge distillation to maintain not only the output logit but also its positive and negative components.

RBC [33] is the only prior method that has examined the differences in class distributions between single-step training sets and the complete dataset. During incremental training, it employs pairs of input images, one normal and the other with new-class pixels erased. Additionally, it enhances the weight of old-class pixels in the loss function. These strategies do help mitigate the bias of classifiers towards new classes. However, they do not take into account the actual class proportions within the complete dataset when rectifying the bias, offering no assurance that that the classifier's bias can be adjusted to an appropriate level. In contrast, our approach, which replays old-class prototypes based on the recorded class proportion in all past training samples, holds an advantage.

## 3 Proposed method

### 3.1 Problem definition and basic setup

In CISS, a model incrementally learns among $T$ steps. For each step $t \in \{1, \ldots, T\}$, the training set $D^t$ is consisted of $N^t$ images $\{x_i^t\}_{i=1}^{N^t} \subseteq \mathbb{R}^{H \times W \times 3}$ and corresponding labels $\{y_i^t\}_{i=1}^{N^t} \subseteq \mathbb{R}^{H \times W}$, where $H \times W$ indicates the image size. During step $t$, only a subset of classes, $L^t$, is regarded as foreground classes with ground-truth labels in $y_i^t$. All other pixels, including the actual background, past classes $L^{1:t-1}$, and future classes $L^{t+1:T}$, are labeled as *background*. Note that the sets of foreground classes for different steps are disjoint, *i.e.*, $L^t \cap L^{1:t-1} = \emptyset$ and $L^t \cap L^{t+1:T} = \emptyset$.

The model at step $t$ is denoted as $M^t$, consisting of a feature extractor $\Psi_{\theta^t}$ with parameters $\theta^t$ and classifiers $\Phi_{\omega^t}$ with parameters $\omega^t$. $\Phi_{\omega^t}$ includes the classifier for each past or current foreground class $l \in L^{1:t}$, represented as $\Phi_{\omega_l^t}$. Before training at step $t$, $\theta^t$ and $\{\omega_l^t\}_{l \in L^{1:t-1}}$ are initialized by

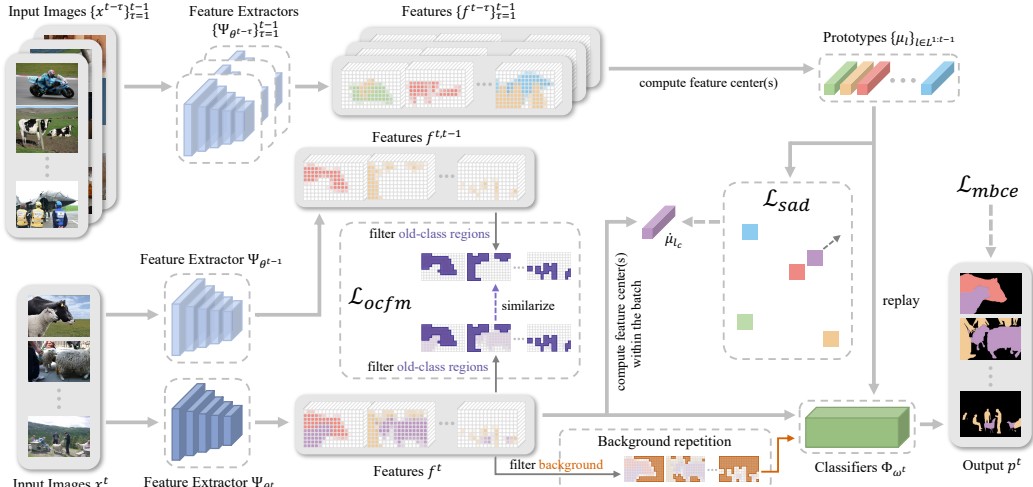

Figure 1: Overview of the proposed STAR. In earlier steps, prototypes for past foreground classes have been saved. At the current step, prototypes are replayed, and background pixels are repeated for classifiers. The OCFM loss maintains similarity between old-class region features from current and previous feature extractors, while the SAD loss differentiates features for similar old-new class pairs.

$\theta^{t-1}$ and $\{\omega_l^{t-1}\}_{l\in L^{1:t-1}}$, and $\{\omega_l^t\}_{l\in L^t}$ is randomly initialized. For an input image $x_i^t$, the output of $M^t$ is denoted as $p_i^t \in \mathbb{R}^{H\times W}$, which can be computed by $p_i^t = M^t(x_i^t) = \Phi_{\omega^t} \circ \Psi_{\theta^t}(x_i^t)$. The extracted features are denoted as $f_i^t \in \mathbb{R}^{h\times w\times c}$, which can be computed by $f_i^t = \Psi_{\theta^t}(x_i^t)$. Here, $h\times w$ and $c$ represent the spatial size and the number of channels, respectively. When feeding a current image $x_i^t$ to a previous model $M^{t-\tau}$ or feature extractor $\Psi_{\theta^{t-\tau}}$, the output is given by $p_i^{t,t-\tau} = M^{t-\tau}(x_i^t)$, and the features are given by $f_i^{t,t-\tau} = \Psi_{\theta^{t-\tau}}(x_i^t)$.

## 3.2 Overview of the method

The overview of the proposed STAR is shown in Fig. 1. After training at each step $t-\tau$, the prototype of each foreground class $l\in L^{t-\tau}$ and essential statistics are computed and saved. At a subsequent step $t$, the input to the classifiers includes not only the features extracted from the input image but also proportionally replayed prototypes of all past foreground classes as well as repetitions of the features in the background region, aligning the sample distribution with the training set accumulated up to this step, $D^{1:t}$. Moreover, we employ the OCFM loss, $\mathcal{L}_{ocfm}$, between the features $f_i^t$ and $f_i^{t,t-1}$ to preserve old-class features and apply the SAD loss, $\mathcal{L}_{sad}$, to differentiate the features between similar old-new class pairs. Consequently, our total loss function is:

$$\mathcal{L} = \mathcal{L}_{mbce} + \alpha\mathcal{L}_{ocfm} + \beta\mathcal{L}_{sad}. \tag{1}$$

Here, $\mathcal{L}_{mbce}$ denotes a multiple binary cross-entropy (mBCE) loss with a positive weight of $\gamma$ as in [1, 3], which is proved to be more suitable for CISS than a multi-class cross-entropy loss. $\mathcal{L}_{ocfm}$ and $\mathcal{L}_{sad}$ denote the OCFM and SAD losses, respectively, balanced by hyperparameters $\alpha$ and $\beta$.

## 3.3 Prototype preservation and replay

As previously stated, in a single-step training set, the proportion of pixels from current foreground classes is excessively high, resulting in biased predictions. To address this issue, we aim to comprehensively reconstruct the class distribution of training samples at each step $t$, aligning it with the distribution in the union of all training sets up to this step $D^{1:t}$. This distribution represents the achievable limit, as it is not possible to know the data from future steps in practice, and it should also be close to the distribution in the complete dataset accumulated up to this step. The most straightforward solution to this problem is to save and replay all past images, but this does not meet the requirements of continual learning, necessitating an alternative solution.

We argue that the feature extractor is relatively generalized in extracting features and does not contain information strongly correlated with the classes. The ability of SSUL [3] to learn new classes even

if the feature extractor parameters are frozen supports this claim. As a result, the network's bias primarily stems from the classifiers. Therefore, we propose creating a prototype for each learned foreground class and replaying these prototypes in subsequent steps, allowing them to go through the classifiers together with the features extracted from input images. This process can correct the foreground class distribution accessed by the classifiers, eliminating its bias. Meanwhile, we have to ensure that each prototype contains the most representative characteristics of its corresponding foreground class with minimal storage cost, thereby efficiently maximizing the correction for the classifiers.

Specifically, after training at each step $t - \tau$, we calculate the number of pixels from each current foreground class $l \in L^{t-\tau}$ in the single-step training set $D^{t-\tau}$:

$$\eta_l = \sum_{i=1}^{N^{t-\tau}} \sum_{j=1}^{h \times w} \mathbb{1}\{\tilde{y}_{i,j}^{t-\tau} = l\}. \tag{2}$$

Here, $j$ indicates the spatial location, $\mathbb{1}$ denotes the indicator function. $\tilde{y}_i^{t-\tau} \in \mathbb{R}^{h \times w}$ represents the downsampled label aligned with the spatial size of the features $f_i^{t-\tau}$ that fed into the classifiers. It is given by $\tilde{y}_i^{t-\tau} = DS(y_i^{t-\tau})$, where $DS$ denotes a downsampling operation. Since each class is treated as a foreground class in only one step, $\eta_l$ is the count of pixels from class $l$ appearing as foreground throughout the entire dataset, which determines the frequency of replaying the prototype of class $l$ in subsequent steps. Still, due to the possible presence of a small number of $l$ pixels appearing in other steps' training sets as background, we utilize the previous model to estimate the number of $l$ pixels in the current training set to fine-tune $\eta_l$. Yet, as this aspect has minimal influence on the final outcomes, we deem it optional and provide clarification in the supplementary material.

Next, in order to build replay samples of class $l$ in future steps, we calculate the feature center of $l$ to capture its primary characteristics by:

$$\mu_l = \frac{\sum_{i=1}^{N^{t-\tau}} \sum_{j=1}^{h \times w} (\hat{f}_{i,j}^{t-\tau} \times \mathbb{1}\{\tilde{y}_{i,j}^{t-\tau} = l\})}{\left\| \sum_{i=1}^{N^{t-\tau}} \sum_{j=1}^{h \times w} (\hat{f}_{i,j}^{t-\tau} \times \mathbb{1}\{\tilde{y}_{i,j}^{t-\tau} = l\}) \right\|_2} \in \mathbb{R}^c, \tag{3}$$

where $\otimes$ denotes the element-wise multiplication, $\|\cdot\|_2$ computes the L2-norm, and $\hat{f}_{i,j}^{t-\tau}$ refers to the L2-normalized feature vector given by $\hat{f}_{i,j}^{t-\tau} = \frac{f_{i,j}^{t-\tau}}{\|f_{i,j}^{t-\tau}\|_2}$. $\mu_l$ indicates the mean direction of all feature vectors for $l$, serving as a prototype of it that embodies the typical characteristic. However, relying solely on this prototype during the replay in subsequent steps may result in inadequately robust training outcomes, as a prototype only provides a single representation of class features without encompassing any variability. Therefore, we also compute the standard deviation of the feature vectors to introduce noise during the replay process. The calculation is given by:

$$\sigma_l = \sqrt{\frac{1}{\eta_l} \sum_{i=1}^{N^{t-\tau}} \sum_{j=1}^{h \times w} \left[ (\hat{f}_{i,j}^{t-\tau} - \mu_l)^2 \times \mathbb{1}\{\tilde{y}_{i,j}^{t-\tau} = l\} \right]} \in \mathbb{R}^c. \tag{4}$$

In addition, since we remove the length information of feature vectors through L2-normalization, we need to restore it during replay. Hence, we also store the mean and standard deviation of the L2-norms of all features for class $l$, denoted as $\mu_{l,norm}$ and $\sigma_{l,norm}$, which are formulated by:

$$\mu_{l,norm} = \frac{1}{\eta_l} \sum_{i=1}^{N^{t-\tau}} \sum_{j=1}^{h \times w} (\left\| \hat{f}_{i,j}^{t-\tau} \right\|_2 \times \mathbb{1}\{\tilde{y}_{i,j}^{t-\tau} = l\}) \in \mathbb{R}^1, \tag{5}$$

$$\sigma_{l,norm} = \sqrt{\frac{1}{\eta_l} \sum_{i=1}^{N^{t-\tau}} \sum_{j=1}^{h \times w} \left[ (\left\| \hat{f}_{i,j}^{t-\tau} \right\|_2 - \mu_{l,norm})^2 \times \mathbb{1}\{\tilde{y}_{i,j}^{t-\tau} = l\} \right]} \in \mathbb{R}^1. \tag{6}$$

With $\eta_l$, $\mu_l$, $\sigma_l$, $\mu_{l,norm}$ and $\sigma_{l,norm}$ saved, robust replay in future steps will be available.

At a subsequent step $t$, the classifier would originally be fed only the features extracted from the input image, $f_i^t$. To correct the ratio between old and new foreground classes, we also input replayed old

class samples into the classifier. For an old class $l \in L^{1:t-1}$, we use a random variable $r_l$ to represent the replayed sample, which is the product of two Gaussian-distributed random variables, the feature vector direction $\kappa \sim \mathcal{G}(\mu_l, \sigma_l{}^2)$ and the L2-norm $\lambda \sim \mathcal{G}(\mu_{l,norm}, \sigma_{l,norm}{}^2)$:

$$r_l = \kappa \times \lambda. \tag{7}$$

For each old class $l$, we replay $r_l$ for $\eta_l$ times per training epoch, which represents the number of pixels from $l$ that appear outside $D^t$. In each epoch, all foreground class pixels in $D^t$ also appear once, so replaying $\eta_l$ times can align the ratio of old and new foreground class samples with the ratio in $D^{1:t}$. Assuming each epoch contains $e$ iterations, we distribute the $\eta_l$ times evenly among these iterations. This implies sampling $r_l$ for $\frac{\eta_l}{e}$ times in each iteration and concatenating these random samples with $f_i^t$ to feed the classifiers for calculating the total gradient to update parameters. In this way, the bias towards any foreground classes is eliminated. Simultaneously, as each sampling of $r_l$ yields different features, and the distribution is controlled by stored standard deviations, the classifiers can capture sufficient potential characteristics of old classes, achieving robust discriminability.

## 3.4 Background repetition

Apart from the ratio between samples of old and new foreground classes, we have to also restore the proportion of background samples to fully rectify the distribution. However, due to the richer semantics contained in the background and the background shift across different steps, condensing its meaning through a single prototype is impossible. Therefore, we choose to repeat background pixels in the current training set $D^t$ with a proper frequency to restore their proportion.

Similar to the strategy in Sec. 3.3, to control the number of background repetitions in future steps, we calculate the number of appeared background pixels at each step $t - \tau$. As numerous background pixels exist in all steps, we not only count the number of background pixels in the single-step training set but also add the background pixel counts saved in previous steps, given by:

$$\eta_{bg}^{t-\tau} = \begin{cases} \sum_{i=1}^{N^{t-\tau}} \sum_{j=1}^{h \times w} \mathbb{1}\{\tilde{y}_{i,j}^{t-\tau} = 0\} & \text{if } t - \tau = 1 \\ \eta_{bg}^{t-\tau-1} + \sum_{i=1}^{N^{t-\tau}} \sum_{j=1}^{h \times w} \mathbb{1}\{\tilde{y}_{i,j}^{t-\tau} = 0\} & \text{if } t - \tau > 1, \end{cases} \tag{8}$$

where $0$ in $\tilde{y}_{i,j}^{t-\tau}$ indicates the label *background*. In fact, since there may be overlapping background pixels between the training sets in different steps, we scale down $\eta_{bg}^{t-\tau}$ based on some prior assumptions. However, similar to the case of foreground class counts, the impact is minimal, so it is optional and described in the supplementary material.

At each incremental step $t$, we restore the proportion of the background by repeating background pixels in the current training set $D^t$ by a certain time. Specifically, we consider a pixel in $f_i^t$ as a background pixel if it meets two conditions: 1) its label is *background*, and 2) the model at the last step $M^{t-1}$ also predicts it as background. Ideally, these requirements ensure the pixel is neither a current nor past foreground pixel and can be currently categorized as background. Using these criteria, we obtain the set of background pixels $B^t$:

$$B^t = \{f_{i,j}^t \mid \tilde{y}_{i,j}^t = 0 \text{ and } \tilde{p}_{i,j}^{t,t-1} = 0\}, \tag{9}$$

where $\tilde{p}_{i,j}^{t,t-1} = DS(p_{i,j}^{t,t-1})$ is the downsampled prediction of $M^{t-1}$. During training, we repeat pixels in $B^t$ by $\frac{\eta_{bg}^t}{|B^t|}$ times per epoch, where $|\cdot|$ denotes set cardinality, to integrate $\eta_{bg}^t$ additional background pixels, with $\eta_{bg}^t$ representing the count of background pixels outside $D^t$. This aligns the background pixel proportion with that in $D^{1:t}$. By merging prototype replay and background repetition strategies, we rectify the distribution of both foreground classes and background in single-step training samples for the classifier.

## 3.5 Old-class features maintaining loss

The prototype replay strategy introduced in Sec. 3.3 can correct the proportion of old-class samples while conserving storage space. However, its feasibility relies on the premise that feature extraction for old-class regions does not change significantly between steps. If this condition is not met, there will be a discrepancy between the old-class features extracted by the current feature extractor and the replayed old-class prototypes, as the prototypes are created from features generated by previous

feature extractors. In this situation, the replay becomes ineffective because the features that the prototypes represent have become outdated. Furthermore, since there are no constraints on parameter updates, the feature extractor may suffer from uncontrolled catastrophic forgetting of old knowledge. Consequently, our goal is to develop a strategy that maintains old-class features, cooperating with our replay strategy and resisting catastrophic forgetting. At the same time, we do not want the network to lose plasticity for learning new classes. By balancing these objectives, we design the OCFM loss, which selectively maintains features of old-class regions in a spatially targeted manner while simultaneously allowing flexibility for learning new classes.

OCFM maintains old-class features by penalizing the changes from the features extracted by the previous feature extractor, $f_i^{t,t-1}$, to the features extracted by the current feature extractor, $f_i^t$, in old-class regions. For a pixel, we consider it belongs to old-class regions by two requirements: 1) its current label is *background*, and 2) the previous model $M^{t-1}$ predicts it as a foreground pixel. For these old-class pixels, the OCFM loss punishes the mean squared error between $f_i^t$ and $f_i^{t,t-1}$ by:

$$
\begin{aligned}
\mathcal{L}_{ocfm} = MSE(f_i^t \times \mathbb{1}\{\tilde{y}_{i,j}^t = 0\} \times \mathbb{1}\{\tilde{p}_{i,j}^{t,t-1} = 0\}, \\
f_i^{t,t-1} \times \mathbb{1}\{\tilde{y}_{i,j}^t = 0\} \times \mathbb{1}\{\tilde{p}_{i,j}^{t,t-1} = 0\}),
\end{aligned}
\tag{10}
$$

in which $MSE$ signifies the mean squared error.

Compared to distillations that operate on the entire spatial region [1, 9, 31], our OCFM loss specifically targets old-class regions, preserving these features to collaborate with our replay strategy and combat catastrophic forgetting. Simultaneously, the features in other regions can be updated without constraints, ensuring that the flexibility to learn new classes remains uncompromised.

### 3.6 Similarity-aware discriminative loss

The previous sections address the bias resulting from distribution differences between single-step training sets and the complete dataset. Despite the bias being eliminated, some similar class pairs may still be prone to confusion, as the feature extractor might not generate sufficiently discriminative features for them. When these similar classes are trained in the same step, the situation may be less severe since the mBCE loss drives their features to exhibit adequate differences. However, when they are trained in different steps, the mBCE loss alone is not enough to encourage the feature extractor to generate discriminative features. This is because the old-class features are constrained by the OCFM loss, and most old-class representations that eventually flow to the mBCE loss are derived from prototypes, which, although close to the authentic features, are slightly inferior in richness.

To address this problem, a straightforward solution is to introduce a differentiation loss to all old-new class feature center pairs, seeking to maximize their divergence. However, this lacks targeting, as some feature pairs are already widely separated, and we should focus on those similar pairs. To this end, we refine this strategy to form our SAD loss: for each current class, we only penalize the distance between its feature center and the nearest old-class feature center, thereby directing the differentiation effort to where it is most needed. At step $t$, the feature center of an old class $l_o \in L^{1:t-1}$ is the previously saved prototype, $\mu_{l_o}$, and the feature center of a current class $l_c \in L^t$ is computed within the batch being processed, given by:

$$
\dot{\mu}_{l_c} = \frac{\sum_{i \in U} \sum_{j=1}^{h \times w} (\hat{f}_{i,j}^t \times \mathbb{1}\{\tilde{y}_{i,j}^t = l\})}{\left\| \sum_{i \in U} \sum_{j=1}^{h \times w} (\hat{f}_{i,j}^t \times \mathbb{1}\{\tilde{y}_{i,j}^t = l\}) \right\|_2},
\tag{11}
$$

where $U$ represents the set of indices of images belonging to the current batch. After computing the feature centers for all current foreground classes, we can achieve the SAD loss as:

$$
\mathcal{L}_{sad} = \frac{1}{|L^t|} \sum_{l_c \in L^t} \min_{l_o \in L^{1:t-1}} \|\mu_{l_o} - \dot{\mu}_{l_c}\|_2.
\tag{12}
$$

With $\mathcal{L}_{sad}$, the features of each new class will be distanced from the most similar old-class prototype. When the distance between a new-class feature center and its closest old-class prototype is far enough, surpassing the distance to another old-class prototype, that old-class becomes the new closest one, and $\mathcal{L}_{sad}$ turns to penalize the distance between this new pair. Ultimately, the features of the new class will be far enough away from all old class prototypes.

Table 1: Quantitative comparison on Pascal VOC 2012 between our STAR and previous non-replay methods (top half) and replay-based methods (bottom half) under the *overlapped* setting.

| Model | 19-1 (2 steps) | | | 15-5 (2 steps) | | | 15-1 (6 steps) | | | 10-1 (11 steps) | | | 5-3 (6 steps) | | |
|---|---|---|---|---|---|---|---|---|---|---|---|---|---|---|---|
| | old | new | all | old | new | all | old | new | all | old | new | all | old | new | all |
| MiB [2] | 70.2 | 22.1 | 67.8 | 75.5 | 49.4 | 69.0 | 35.1 | 13.5 | 29.7 | 12.3 | 13.1 | 12.7 | 57.1 | 42.6 | 46.7 |
| SDR [23] | 69.1 | 32.6 | 67.4 | 75.4 | 52.6 | 69.9 | 44.7 | 21.8 | 39.2 | - | - | - | 17.5 | 19.2 | 18.7 |
| PLOP [9] | 75.4 | 37.4 | 73.5 | 75.7 | 51.7 | 70.1 | 65.1 | 21.1 | 54.6 | 44.0 | 15.5 | 30.5 | | | |
| SSUL [3] | 77.7 | 29.7 | 75.4 | 77.8 | 50.1 | 71.2 | 77.3 | 36.6 | 67.6 | 71.3 | 46.0 | 59.3 | 72.4 | 50.7 | 56.9 |
| RCIL [31] | - | - | - | 78.8 | 52.0 | 72.4 | 70.6 | 23.7 | 59.4 | 55.4 | 15.1 | 34.3 | - | - | - |
| RBC [33] | 77.3 | 55.6 | 76.2 | 76.6 | 52.8 | 70.9 | 69.5 | 38.4 | 62.1 | - | - | - | 69.6 | 53.5 | 58.1 |
| DKD [1] | 77.8 | 41.5 | 76.0 | 78.8 | 58.2 | 73.9 | 78.1 | 42.7 | 69.7 | 73.1 | 46.5 | 60.4 | - | - | - |
| UCD [29] | 75.9 | 39.5 | 74.0 | 75.0 | 51.8 | 69.2 | 66.3 | 21.6 | 55.1 | 42.3 | 28.3 | 35.3 | - | - | - |
| AWT [12] | - | - | - | 78.0 | 50.2 | 71.4 | 77.0 | 37.6 | 67.6 | 73.1 | 47.0 | 60.7 | 71.6 | 51.4 | 57.1 |
| Ours | 78.0 | 47.1 | **76.5** | 79.5 | 58.9 | **74.6** | 79.5 | 50.6 | **72.6** | 73.1 | 55.4 | **64.7** | 71.9 | 61.5 | **64.4** |
| RECALL [20] | 68.1 | 55.3 | 68.6 | 67.7 | 54.3 | 65.6 | 67.8 | 50.9 | 64.8 | 65.0 | 53.7 | 60.7 | - | - | - |
| SSUL-M(100) [3] | 77.8 | 49.8 | 76.5 | 78.4 | 55.8 | 73.0 | 78.4 | 49.0 | 71.4 | 74.0 | 53.2 | 64.1 | 71.3 | 53.2 | 58.4 |
| DKD-M(100) [1] | 78.0 | 57.7 | **77.0** | 79.1 | 60.6 | 74.7 | 78.8 | 52.4 | 72.5 | 74.0 | 56.7 | 65.8 | 69.8 | 60.2 | 62.9 |
| Ours-M(50) | 77.8 | 56.4 | 76.8 | 79.7 | 61.8 | **75.4** | 79.5 | 55.6 | **73.8** | 74.3 | 57.9 | **66.5** | 71.9 | 62.9 | 65.5 |

Table 2: Quantitative results on ADE20K under the *overlapped* setting.

| Model | 100-50 (2 steps) | | | 100-10 (6 steps) | | | 50-50 (3 steps) | | |
|---|---|---|---|---|---|---|---|---|---|
| | old | new | all | old | new | all | old | new | all |
| MiB [2] | 37.9 | 27.9 | 34.6 | 31.8 | 14.1 | 25.9 | 35.5 | 22.9 | 27.0 |
| PLOP [9] | 41.9 | 14.9 | 32.9 | 40.5 | 13.6 | 31.6 | 48.8 | 21.0 | 30.4 |
| RCIL [31] | 42.3 | 18.8 | 34.5 | 39.3 | 17.6 | 32.1 | 48.3 | 25.0 | 32.5 |
| RBC [33] | 42.9 | 21.5 | 35.8 | 39.0 | 21.7 | 33.3 | 49.6 | 26.3 | 34.2 |
| UCD [29] | 42.1 | 15.8 | 33.3 | 40.8 | 15.2 | 32.3 | 47.1 | 24.1 | 31.8 |
| AWT [12] | 40.9 | 24.7 | 35.6 | 39.1 | 21.3 | 33.2 | 46.6 | 26.9 | 33.5 |
| SSUL-M(300) [3] | 42.8 | 17.5 | 34.4 | 42.9 | 17.7 | 34.5 | 49.1 | 20.1 | 29.8 |
| DKD-M(300) [1] | 42.4 | 23.0 | 36.0 | 41.7 | 20.1 | 34.6 | 48.8 | 26.3 | 33.9 |
| Ours | 42.4 | 24.2 | **36.4** | 42.0 | 20.6 | **34.9** | 48.7 | 27.2 | **34.4** |

## 4 Experiments

### 4.1 Experimental setups

**Datasets and evaluation metric.** We evaluate our method on two common public datasets: Pascal VOC 2012 [11] and ADE20K [34]. Pascal VOC 2012 comprises 10,582 training images and 1,449 validation images, covering 20 classes. ADE20K includes 20,210 training images and 2,000 validation images, spanning 150 classes. Following previous works [1, 2, 3], we use mean Intersection-over-Union (mIoU) as our evaluation metric, representing the average IoU across all classes.

**Experimental protocols.** Following previous works, we evaluate our model with various class splits among multiple steps. Each split is denoted as $N_1$-$N_2$, where $N_1$ and $N_2$ refer to the number of classes in the first step and in each incremental step, respectively. We have 19-1, 15-5, 15-1, 10-1, and 5-3 splits for Pascal VOC 2012, and 100-50, 100-10, and 50-50 splits for ADE20K. Furthermore, we consider two CISS settings: *disjoint* and *overlapped*. The difference between them is that the *disjoint* excludes images with future class pixels in $D^t$, while the *overlapped* includes them (future class pixels are labeled as *background*). Since the *overlapped* is closer to real-world scenarios, we report the results of it in this paper and put the results of the *disjoint* in the supplementary material.

**Implementation details.** Following [1, 2, 9], we use DeepLabv3 [5] with a ResNet-101 [13] backbone pre-trained on ImageNet [7] as our segmentation network. In line with [1, 2, 3, 9], training strategies differ for the two datasets. For Pascal VOC 2012, we train for 60 epochs with an initial learning rate of 0.001 for the first step and 0.0001 for incremental steps, and empirically set $\gamma$ to 4. For ADE20K, we train for 100 epochs with an initial learning rate of 0.00025 for the first step and 0.000025 for incremental steps, and set $\gamma$ to 30. For both datasets, we use an SGD optimizer with a momentum of 0.9. The batch size is set to 24, and $\alpha$ and $\beta$ are set to 5 and 0.05, respectively. We implement our model using PyTorch [25] and employ two NVIDIA RTX 3090 GPUs for acceleration. We have also implemented our network using the MindSpore Lite tool[1].

---

[1] 

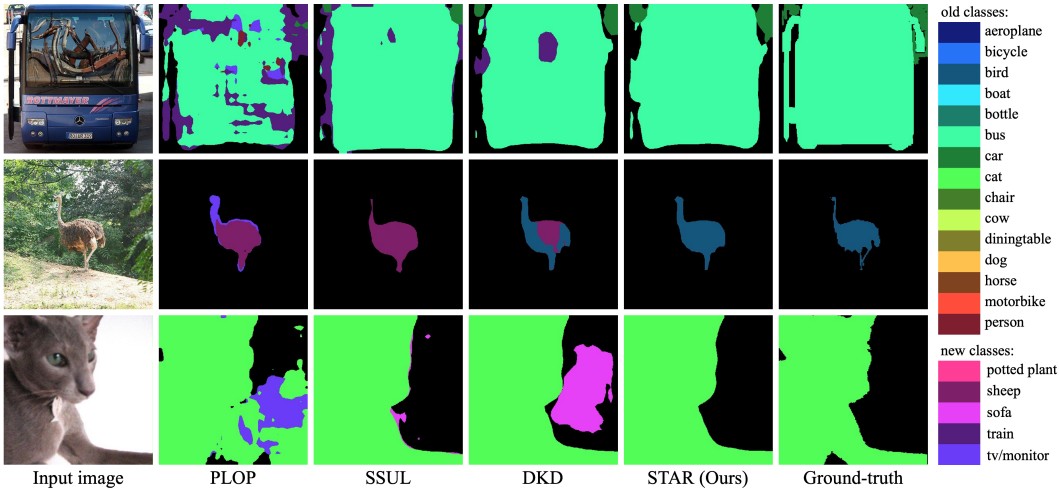

| Input image | PLOP | SSUL | DKD | STAR (Ours) | Ground-truth |

old classes:
- aeroplane
- bicycle
- bird
- boat
- bottle
- bus
- car
- cat
- chair
- cow
- diningtable
- dog
- horse
- motorbike
- person

new classes:
- potted plant
- sheep
- sofa
- train
- tv/monitor

Figure 2: Qualitative comparison on Pascal VOC 2012 between STAR and previous methods.

## 4.2 Comparisons

We compare STAR with previous CISS methods both quantitatively and qualitatively. Among them, MiB [2], SDR [23], PLOP [9], SSUL [3], RCIL [31], RBC [33], DKD [1], UCD [29], and AWT [12] are non-replay-based, while RECALL [20], SSUL-M [3], and DKD-M [1] are replay-based methods.

**Quantitative comparison.** The quantitative comparison results for Pascal VOC 2012 and ADE20K datasets are presented in Table 1 and Table 2, respectively, where "old" indicates the mIoU of classes in the first step while "new" indicates the mIoU of incremental classes. Compared to previous non-replay-based models, our STAR consistently achieves leading performance across various class splits. Particularly in more challenging 15-1, 10-1, and 5-3 splits that contain more steps, our method outperforms others by at least 2.9, 4.0, and 6.3 mIoU, respectively. Notably, although STAR requires additional storage space for prototypes and statistics, the space needed is minimal (~41KB), causing no significant unfairness compared to non-replay-based models. Meanwhile, even compared with other replay-based models, STAR remains highly competitive, achieving performance on par with the best-performing DKD-M while requiring less than 0.5% of its storage space (DKD-M cost ~10MB to save 100 raw images from past steps). If we add 50 extra raw images to provide more realistic features (Ours-M), the performance of our model can be further improved, surpassing DKD-M in four out of five class splits. Remarkably, even in this scenario, the required storage is just about half of that needed by DKD-M.

On the ADE20K dataset, our model performs even better, outperforming all previous non-replay-based and replay-based methods without storing any extra raw images. Despite the larger class count on ADE20K necessitating the storage of more prototypes, the required storage remains a mere ~2% of that demanded by DKD-M and SSUL-M, both of which store 300 raw images.

In summary, our model achieves outstanding performance while requiring, on average, only ~1% of the storage needed by other replay-based models, and it requires no extra image beyond the datasets. Moreover, privacy concerns tied to storing raw images are avoided. These demonstrates the significance of our approach to reconstruct class distribution through prototype replay.

**Qualitative comparison.** Fig. 2 provides a qualitative comparison using the challenging *overlapped* 15-1 protocol on Pascal VOC 2012, highlighting our method's superiority over previous state-of-the-art CISS methods [1, 3, 9]. The prevalent errors in other methods are false positives for new classes. For instance, in the first two examples, some pixels of old classes *bus* and *bird* are identified as new classes *train*, *sheep*, or *tv/monitor*. In the third example, some background pixels are misclassified as new classes *sofa* or *tv/monitor*. Such errors stem from the underrepresentation and limited diversity of old-class or background samples in the single-step training set when training new classes, resulting in a bias towards them. In contrast, our method effectively addresses these problems by rectifying the distribution of single-step training samples and providing new-class classifiers with access to old-class prototypes as negative samples.

Table 3: Ablation study results of our method in the Pascal VOC 2012 dataset.

| *Replay* | *BR* | *OCFM* | *SAD* | overlapped 15-1 (6 steps) | | |
|---|---|---|---|---|---|---|
| | | | | old | new | all |
| | ✓ | ✓ | ✓ | 79.3 | 41.8 | 70.3 |
| ✓ | | ✓ | ✓ | 78.3 | 35.5 | 68.1 |
| ✓ | ✓ | | ✓ | 37.3 | 12.8 | 31.4 |
| ✓ | ✓ | ✓ | | 79.3 | 48.2 | 71.6 |
| ✓ | ✓ | ✓ | ✓ | 79.5 | 50.6 | **72.6** |

## 4.3 Ablation study

Table 3 shows our ablation study results for STAR under the challenging *overlapped* 15-1 protocol, assessing its four core components: prototype replay (*Replay*), background repetition (*BR*), old-class features maintaining loss (*OCFM*), and similarity-aware discriminative loss (*SAD*). Each component is individually removed from the full model to evaluate its contribution to the overall performance.

**Effectiveness of the prototype replay strategy (*Replay*).** The first row of Table 3 presents the results without *Replay*. Compared to the full model's results shown in the last row, it is clear that *Replay* boosts the performance by 2.3 mIoU. Although *Replay* involves replaying old-class prototypes, it primarily enhances the accuracy of new classes, demonstrating an 8.8 mIoU improvement. This effect arises from *Replay* reducing the proportion of new-class samples flowing into the classifiers and providing new-class classifiers with sufficient access to old-class features. This dual effect mitigates classifier bias and lessens the chance of misclassifying non-new-class pixels, especially those old-class pixels, as new-class pixels. Consequently, we observe a significant increase in the mIoU of new classes and a slight improvement in the mIoU of old classes.

**Effectiveness of the background repetition strategy (*BR*).** As displayed in Table 3, excluding *BR* (second row) leads to a performance drop compared to the full model (last row). Specifically, *BR* brings a 4.5 mIoU improvement overall and notably boosts the mIoU of new classes by 15.1. Despite simply repeating background pixels from the current training set, *BR* effectively corrects the classifier bias due to the high proportion and diversity of background pixels, even within a single-step training set. Likewise, because of the high proportion of background pixels, *BR* introduces more additional samples than *Replay*, making its removal lead to a more noticeable performance degradation.

**Effectiveness of the old-class features maintaining loss (*OCFM*).** The significance of *OCFM* can be recognized when comparing the third and final rows of Table 3. Upon removal of *OCFM*, the results experience a complete breakdown, with a dramatic 41.2 mIoU drop compared to the full model. *OCFM* is a crucial mechanism in our model for maintaining prior knowledge. Its absence not only makes replay ineffective but also triggers uncontrolled network parameter updates, leading to intense catastrophic forgetting. Thus, the performance of both new and old classes suffers greatly.

**Effectiveness of the similarity-aware discriminative loss (*SAD*).** The final component under validation is *SAD*, designed to differentiate the features of similar old-new class pairs. The penultimate row in Table 3 illustrates the performance without *SAD*, registering a 1.0 mIoU reduction. This attests to the efficacy of *SAD* in augmenting feature differences between similar old-new class pairs, thereby making them more distinguishable to the classifiers, leading to improved prediction accuracy.

## 5 Conclusion

In this paper, we present a novel class-incremental semantic segmentation (CISS) method named STAR that addresses the bias towards part of classes due to distribution discrepancies between single-step training sets and the complete dataset. STAR incorporates two key strategies: prototype replay and background repetition, aiming to correct the proportions of foreground classes and background, respectively. By saving and replaying prototypes instead of raw images, STAR reduces storage costs to ~1% of other replay-based methods, without requiring extra data. Furthermore, we propose an old-class features maintaining loss to keep old-class features unchanged while preserving the flexibility to learn new classes and a similarity-aware discriminative loss to selectively differentiate features of most similar old-new class pairs, making them distinguishable for the classifiers. Comparisons with previous state-of-the-art CISS methods and our ablation study demonstrate the superiority of our overall design and the importance of each individual component in our model.

## Acknowledgements

This work was supported in part by the National Key R&D Program of China under Grant 2021ZD0112100, in part by the Hong Kong Innovation and Technology Commission (InnoHK Project CIMDA), in part by the Hong Kong GRF-RGC General Research Fund under Grant 11209819 and Grant 11203820, in part by the National Natural Science Foundation of China under Grant 62002014, in part by the Taishan Scholar Project of Shandong Province under Grant tsqn202306079, in part by Young Elite Scientist Sponsorship Program by the China Association for Science and Technology under Grant 2020QNRC001, in part by CAAI-Huawei MindSpore Open Fund.

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
