# The Supplementary Material for
# Saving 100x Storage: Prototype Replay for Reconstructing Training Sample Distribution in Class-Incremental Semantic Segmentation

**Jinpeng Chen[1], Runmin Cong[2][*][‡], Yuxuan Luo[1], Horace Ho Shing Ip[1][†], and Sam Kwong[3][‡]**
[1]Department of Computer Science, City University of Hong Kong, Hong Kong
[2]School of Control Science and Engineering, Shandong University, Jinan, Shandong, China
[3]Lingnan University, Hong Kong
jinpechen2-c@my.cityu.edu.hk, rmcong@sdu.edu.cn, yuxuanluo4-c@my.cityu.edu.hk,
horace.ip@cityu.edu.hk, samkwong@ln.edu.hk

## 1   Pseudo code description

In order to help readers better understand our STAR, we summarize its workflow in Algorithm 1.

## 2   Pixel count fine-tuning

In the main body of our paper, we discussed the need for fine-tuning the count of past foreground class and background pixels, enabling them to serve as frequencies for replaying prototypes or repeating background pixels. In this section, we provide further elaboration on this fine-tuning method. This method is somewhat intricate and has a minimal impact on the results (as proved in Sec. 3.3), so we deem it optional. Nevertheless, it is maintained in our default configuration to ensure the logical integrity.

### 2.1   The count of foreground class pixels

In the main body of our paper, we discussed how a random sample of a past foreground class $l \in L^{t-\tau}$, denoted as $r_l$, is replayed $\eta_l$ times per training epoch at each subsequent step $t$. The goal of this strategy is to equalize the count of $l$ pixels in the current single-step training samples with that in $D^{1:t}$, which represents the union of all training sets up to the present. As $D^t$ is the current training set, we need to compensate for the number of $l$ pixels within the set $D^{1:t} - D^t$. This compensatory count, however, does not precisely match $\eta_l$, which is the count of $l$ pixels in $D^{t-\tau}$ where $l$ is considered as foreground. The discrepancy arises due to the overlap between training sets at different steps, and the likelihood that pixels from $l$ might appear as background in the training sets of other steps. Consequently, fine-tuning of $\eta_l$ is required.

Assuming $l$ is a foreground class at step $t - \tau$, we redefine $\eta_l$ as $\eta_l^{t-\tau}$ within this section. At each subsequent step $t$, we will perform fine-tuning on $\eta_l^{t-\tau}$, resulting in $\eta_l^t$.

**Fine-tuning in the overlapped setting.** In the *overlapped* setting, overlap exists between the training sets at different steps. Therefore, a pixel belonging to $l$, which is treated as a foreground class at step

---

[*]Runmin Cong is also with the Key Laboratory of Machine Intelligence and System Control, Ministry of Education, Jinan, Shandong, China.

[†]Horace Ho Shing Ip is also with the Centre for Innovative Applications of Internet and Multimedia Technologies, City University of Hong Kong, Hong Kong.

[‡]Corresponding authors.

37th Conference on Neural Information Processing Systems (NeurIPS 2023).

**Algorithm 1:** Pseudo code of STAR

---

**Require:** Feature extractor $\Psi_\theta$, classifiers $\Phi_\omega$, number of steps $T$, number of epochs $N_{epoch}$, training sets for all steps $\{D^t\}_{t=1}^T$, foreground classes for all steps $\{L^t\}_{t=1}^T$

**for** $t$ in 1 to $T$ **do**

    **if** $t = 1$ **then**
        | Randomly initialize $\theta, \omega$
    **else**
        Initialize $\theta$, $\{\omega_l\}_{l \in L^{1:t-1}}$ with $\theta^{t-1}$, $\{\omega_l^{t-1}\}_{l \in L^{1:t-1}}$
        Randomly initialize $\{\omega_l\}_{l \in L^t}$
    **end**

    // After initialization, $\theta$ and $\omega$ are re-denoted as $\theta^t$ and $\omega^t$

    **for** $n_{epoch}$ in 1 to $N_{epoch}$ **do**
        **for** each batch $U$ in $D^t$ **do**
            $\Psi_{\theta^t}$ extract features, achieving $f^t$
            **if** $t = 1$ **then**
                Feed $f^t$ to $\Phi_{\omega^t}$, achieving the output $p^t$
                // mBCE loss
                Compute $L_{mbce}$ as the multiple binary cross-entropy between $p^t$ and the ground-truth
                // parameter update
                Compute the gradient for $\mathcal{L}_{mbce}$, then update $\theta^t$ and $\omega^t$
            **else**
                $\Psi_{\theta^{t-1}}$ extract features, producing $f^{t,t-1}$
                Feed $f^{t,t-1}$ to $\Phi_{\omega^{t-1}}$, achieving the prediction of old classes, $p^{t,t-1}$
                // prototype replay
                Construct Gaussian-distributed random variables $\{r_l\}_{l \in L^{1:t-1}}$ based on saved prototypes and statistics
                Sample each $r_l$ a certain number of times based on the saved occurrence count of $l$ pixels and concatenate all random samples together
                // background repetition
                Identify background regions by excluding pixels currently labeled as foreground class and old-class pixels estimated in $p^{t,t-1}$
                Duplicate background pixels a certain number of times based on the saved occurrence count of background pixels and concatenate all duplications together
                // mBCE loss
                Concatenate aforementioned random samples and background duplications with $f^t$, feed them to $\Phi_{\omega^t}$, achieving the output $p^t$
                Compute $\mathcal{L}_{mbce}$ as the multiple binary cross-entropy between $p^t$ and corresponding ground-truths
                // old-class features maintaining loss
                Identify old-class pixels with the help of $p^{t,t-1}$
                Compute $\mathcal{L}_{ocfm}$ as the mean square error between $f^t$ and $f^{t,t-1}$ in old-class regions
                // similarity-aware discriminative loss
                Compute $\mathcal{L}_{sad}$ as the mean of the minimum distance from the feature center of each current foreground class (computed within the current batch) to all saved old-class prototypes
                // parameter update
                Compute the gradient for $\mathcal{L}_{mbce} + \alpha\mathcal{L}_{ocfm} + \beta\mathcal{L}_{sad}$, then update $\theta^t$ and $\omega^t$
            **end**
        **end**
    **end**

    Compute and save the occurrence counts for all current foreground class pixels and background pixels
    Compute and save the prototypes and statistics of pixel-level features for all current foreground classes

**end**

**return** $\Psi_{\theta^T}$ and $\Phi_{\omega^T}$

---

$t - \tau$, might appear as background at a subsequent step $t$. However, within the *overlapped* setting, all the pixels from class $l$ throughout the entire training set $D^{1:T}$ have already been present as foreground in $D^{t-\tau}$, so the $l$ pixels in $D^t$ are duplicated. Therefore, we should subtract the count of such pixels from $\eta_l^{t-\tau}$, ensuring that the total number of $l$ samples input to the classifiers is consistent with that in $D^{1:t}$. While it is impossible to determine the exact number of such pixels because they are labeled as *background*, we can leverage the prediction of the preceding model, $\tilde{p}_{i,j}^{t,t-1}$, to approximate it. Hence, the updated count $\eta_l^t$ can be computed by:

$$\eta_l^t = \eta_l^{t-\tau} - \sum_{i=1}^{N^t} \sum_{j=1}^{h \times w} \mathbb{1}\{\tilde{p}_{i,j}^{t,t-1} = l\}. \tag{1}$$

Given that $M^{t-1}$ has already been trained to distinguish class $l$, it should ideally yield a reliable count. Therefore, by incorporating $r_l$ for $\eta_l^t$ times at step $t$, we can compensate for the $l$ pixels within the set $D^{1:t} - D^t$.

**Fine-tuning in the disjoint setting.** In the *disjoint* setting, the situation diverges as there is no overlap between different single-step training sets. Given that the current training set does not contain images with future-class pixels, $l$ pixels that coexist in images with future-class pixels will not be present in $D^{t-\tau}$. Consequently, unlike in the *overlapped* setting, the count of $l$ pixels in $D^{1:t} - D^t$ may surpass $\eta_l^{t-\tau}$ in the *disjoint* setting. This discrepancy necessitates an increase in the replay frequency of $r_l$ beyond $\eta_l^{t-\tau}$. Specifically, considering that the training set of each subsequent step $t$ may include $l$ pixels, we progressively update the count as follows:

$$\eta_l^t = \eta_l^{t-1} + \sum_{i=1}^{N^t} \sum_{j=1}^{h \times w} \mathbb{1}\{\tilde{p}_{i,j}^{t,t-1} = l\}, \tag{2}$$

which is also based on the predictions of the preceding model. Ideally, it approximates $\eta_l^t$ to the count of $l$ pixels within $D^{1:t}$ (and approximates $\eta_l^{t-1}$ to the count within $D^{1:t-1} = D^{1:t} - D^t$). Therefore, at step $t$, we replay $r_l$ per training epoch at a frequency of $\eta_l^{t-1}$, rather than $\eta_l^t$.

## 2.2 The count of background pixels

Similar to the count of foreground class pixels, the count of background pixels also requires fine-tuning. Two main factors necessitate this adjustment: 1) A minor proportion of past background pixels might belong to the current foreground classes, warranting a subtraction of this proportion. 2) Training sets at different steps may have some overlap, requiring the deduction of duplicated background pixels. It is noteworthy that these two factors are only pertinent in the *overlapped* setting. In the *disjoint* setting, where past images do not possibly contain current foreground classes and no overlap exists between the training sets at different steps, no adjustment is needed.

In the main body of our paper, the count of background pixels up to step $t - \tau$ is expressed as follows:

$$\eta_{bg}^{t-\tau} = \begin{cases} \sum_{i=1}^{N^{t-\tau}} \sum_{j=1}^{h \times w} \mathbb{1}\{\tilde{y}_{i,j}^{t-\tau} = 0\} & \text{if } t - \tau = 1 \\ \eta_{bg}^{t-\tau-1} + \sum_{i=1}^{N^{t-\tau}} \sum_{j=1}^{h \times w} \mathbb{1}\{\tilde{y}_{i,j}^{t-\tau} = 0\} & \text{if } t - \tau > 1. \end{cases} \tag{3}$$

Considering the factors mentioned above, we adjust the formula in the *overlapped* setting to:

$$\eta_{bg}^{t-\tau} = \begin{cases} \sum_{i=1}^{N^{t-\tau}} \sum_{j=1}^{h \times w} \mathbb{1}\{\tilde{y}_{i,j}^{t-\tau} = 0\} & \text{if } t - \tau = 1 \\ (1 - \delta|L^{t-\tau}|)\eta_{bg}^{t-\tau-1} + (1 - \epsilon) \sum_{i=1}^{N^{t-\tau}} \sum_{j=1}^{h \times w} \mathbb{1}\{\tilde{y}_{i,j}^{t-\tau} = 0\} & \text{if } t - \tau > 1, \end{cases} \tag{4}$$

where $\delta$ and $\epsilon$ are two hyper-parameters. $\delta$ represents the proportion of previous background pixels that belong to each current foreground class, which is empirically set to $0.01$. $\epsilon$ represents the proportion of background pixels in the current training set that overlap with previous background pixels, which is empirically set to $0.5$. These two adjustments correspond to the two factors described above. Except for this formula, all other parts of the background repetition are the same as stated in the main body.

Table 1: Quantitative comparison on Pascal VOC 2012 between our STAR and previous non-replay methods (top half) and replay-based methods (bottom half) under the *disjoint* setting.

| Model | 19-1 (2 steps) | | | 15-5 (2 steps) | | | 15-1 (6 steps) | | |
|---|---|---|---|---|---|---|---|---|---|
| | old | new | all | old | new | all | old | new | all |
| MiB [2] | 69.6 | 25.6 | 67.4 | 71.8 | 43.3 | 64.7 | 46.2 | 12.9 | 37.9 |
| SDR [7] | 69.9 | 37.3 | 68.4 | 73.5 | 47.3 | 67.2 | 59.2 | 12.9 | 48.1 |
| SSUL [3] | 77.4 | 22.4 | 74.8 | 76.4 | 45.6 | 69.1 | 74.0 | 32.2 | 64.0 |
| RCIL [9] | - | - | - | 75.0 | 42.8 | 67.3 | 66.1 | 18.2 | 54.7 |
| RBC [10] | 76.4 | 45.8 | 75.0 | 75.1 | 49.7 | 69.9 | 61.7 | 19.5 | 51.6 |
| DKD [1] | 77.4 | 43.6 | 75.8 | 77.6 | 54.1 | 72.0 | 76.3 | 39.4 | 67.5 |
| UCD [8] | 75.7 | 31.8 | 73.5 | 67.0 | 39.3 | 60.1 | 50.8 | 13.3 | 41.4 |
| Ours | 77.9 | 43.4 | **76.2** | 78.4 | 57.4 | **73.4** | 78.1 | 46.6 | **70.6** |
| RECALL [5] | 65.0 | 47.1 | 65.4 | 69.2 | 52.9 | 66.3 | 67.6 | 49.2 | 64.3 |
| SSUL-M(100) [3] | 77.6 | 43.9 | 76.0 | 76.5 | 48.6 | 69.8 | 76.5 | 43.4 | 68.6 |
| DKD-M(100) [1] | 77.6 | 56.9 | 76.6 | 77.7 | 55.4 | 72.4 | 77.3 | 48.2 | 70.3 |
| Ours-M(50) | 77.8 | 53.7 | **76.7** | 78.4 | 58.6 | **73.7** | 77.8 | 49.0 | **71.0** |

## 3 More experimental results

### 3.1 Quantitative comparison under the disjoint setting

In Table 1, we present a quantitative comparison between our STAR and previous state-of-the-art methods [1, 2, 3, 5, 7, 8, 9, 10] under the *disjoint* setting. Remarkably, without storing any raw images (Ours), our model surpasses the previous best-performing non-replay-based model, DKD, by an average improvement of 1.6 mIoU across all three protocols. Even compared with the previous best-performing replay-based model, DKD-M, our STAR model still slightly exceeds it by a 0.3 mIoU on average, showing a more surprising performance than under the *overlapped* setting. Moreover, when supplied with 50 additional raw images to provide authentic features (Ours-M), our model exhibits a further improvement in performance, outperforming DKD-M in all three protocols with an average lead of 0.7 mIoU. Considering that the storage space needed by Ours and Ours-M is only ~0.5% and ~50% respectively of what is needed by DKD-M, the performance is impressive.

### 3.2 Error Bars

Table 2: Average mIoU across all classes from five runs of our STAR, under the eight Pascal VOC 2012 protocols.

| Model | overlapped | | | | | disjoint | | |
|---|---|---|---|---|---|---|---|---|
| | 19-1 | 15-5 | 15-1 | 10-1 | 5-3 | 19-1 | 15-5 | 15-1 |
| Ours (time 1) | 76.5 | 74.9 | 72.9 | 64.6 | 64.3 | 76.4 | 73.4 | 70.6 |
| Ours (time 2) | 76.5 | 74.8 | 72.7 | 64.9 | 64.4 | 75.9 | 73.4 | 70.8 |
| Ours (time 3) | 76.6 | 74.5 | 72.6 | 64.6 | 64.5 | 76.4 | 73.1 | 70.5 |
| Ours (time 4) | 76.5 | 74.5 | 72.5 | 64.6 | 64.5 | 76.3 | 73.4 | 70.7 |
| Ours (time 5) | 76.5 | 74.3 | 72.2 | 64.7 | 64.4 | 76.2 | 73.5 | 70.6 |
| Ours | 76.5 (±0.1) | 74.6 (±0.2) | 72.6 (±0.3) | 64.7 (±0.1) | 64.4 (±0.1) | 76.2 (±0.2) | 73.4 (±0.1) | 70.6 (±0.1) |
| Ours-M (time 1) | 76.7 | 75.8 | 74.0 | 66.5 | 65.7 | 76.7 | 73.8 | 71.0 |
| Ours-M (time 2) | 76.8 | 75.6 | 73.9 | 66.6 | 65.4 | 76.6 | 73.8 | 71.2 |
| Ours-M (time 3) | 77.0 | 75.3 | 73.9 | 66.4 | 65.5 | 76.7 | 73.5 | 70.8 |
| Ours-M (time 4) | 76.8 | 75.3 | 73.6 | 66.6 | 65.5 | 76.7 | 73.8 | 71.0 |
| Ours-M (time 5) | 76.7 | 75.0 | 73.6 | 66.5 | 65.4 | 76.7 | 73.7 | 70.9 |
| Ours-M | 76.8 (±0.1) | 75.4 (±0.3) | 73.8 (±0.2) | 66.5 (±0.1) | 65.5 (±0.1) | 76.7 (±0.1) | 73.7 (±0.1) | 71.0 (±0.2) |

In Table 2, we display the results from five separate runs of our STAR under the eight protocols of the Pascal VOC 2012 dataset [4]. Notably, our model demonstrates robust consistency, with outcomes of the five runs being tightly clustered. This clearly illustrates that the superior performance of our model is a result of its intrinsic merits, rather than randomness.

Table 3: Ablation study results under the *disjoint* 15-1 protocol of Pascal VOC 2012.

| *Replay* | *BR* | *OCFM* | *SAD* | disjoint 15-1 (6 steps) | | |
|---|---|---|---|---|---|---|
| | | | | old | new | all |
| | ✓ | ✓ | ✓ | 77.9 | 42.1 | 69.4 |
| ✓ | | ✓ | ✓ | 77.6 | 34.0 | 67.2 |
| ✓ | ✓ | | ✓ | 20.7 | 6.8 | 17.4 |
| ✓ | ✓ | ✓ | | 75.7 | 34.6 | 66.0 |
| ✓ | ✓ | ✓ | ✓ | 78.1 | 46.6 | **70.6** |

### 3.3 More ablation study results

**Ablation study under the disjoint 15-1 protocol.** The results of the ablation study under the *disjoint* 15-1 protocol can be found in Table 3. They reveal a similar trend to those under the *overlapped* 15-1 protocol. Nonetheless, one important observation is the more marked impact of our similarity-aware discriminative loss (*SAD*) within the *disjoint* 15-1 protocol. In *disjoint* 15-1, *SAD* leads to an enhancement in the overall mIoU by 4.6, and a boost in the mIoU of new classes by 12.0. Comparatively, the corresponding enhancements within the *overlapped* 15-1 protocol reached only 1.0 and 2.4. This disparity can be attributed to the fact that the training set may contain future-class pixels in the *overlapped* setting. Despite these pixels being labeled as *background*, they empower the network to distinguish between current classes and similar future classes. That means, although the network cannot know what these future classes are, it does understand that they are distinct from all current classes, thus generating distinctive features. Conversely, the training set in the *disjoint* setting lacks any pixels from future classes. As a result, the network, when faced with a class similar to a previous one, is prone to generate features akin to those of the previous class. This leads to the inability of the classifiers to distinguish between the two. In such a scenario, our *SAD* can guide the network to generate distinct features for these similar old-new class pairs, thereby demonstrating more pronounced efficacy.

Table 4: Ablation study results for pixel count fine-tuning.

| *Replay* | *BR* | *OCFM* | *SAD* | *PCF* | overlapped 15-1 (6 steps) | | | disjoint 15-1 (6 steps) | | |
|---|---|---|---|---|---|---|---|---|---|---|
| | | | | | old | new | all | old | new | all |
| ✓ | ✓ | ✓ | ✓ | | 79.5 | 50.7 | **72.6** | 78.1 | 46.6 | **70.6** |
| ✓ | ✓ | ✓ | ✓ | ✓ | 79.5 | 50.6 | **72.6** | 78.1 | 46.6 | **70.6** |

Table 5: Ablation study results between different knowledge distillations.

| *Replay* | *BR* | *KD* | *OCFM* | *SAD* | overlapped 15-1 (6 steps) | | | disjoint 15-1 (6 steps) | | |
|---|---|---|---|---|---|---|---|---|---|---|
| | | | | | old | new | all | old | new | all |
| ✓ | ✓ | ✓ | | ✓ | 78.9 | 45.9 | 71.0 | 76.5 | 35.3 | 66.7 |
| ✓ | ✓ | | ✓ | ✓ | 79.5 | 50.6 | **72.6** | 78.1 | 46.6 | **70.6** |

Table 6: Ablation study results between different discriminative losses.

| *Replay* | *BR* | *OCFM* | *MD* | *SAD* | overlapped 15-1 (6 steps) | | | disjoint 15-1 (6 steps) | | |
|---|---|---|---|---|---|---|---|---|---|---|
| | | | | | old | new | all | old | new | all |
| ✓ | ✓ | ✓ | ✓ | | 79.6 | 48.5 | 72.2 | 76.2 | 35.9 | 66.6 |
| ✓ | ✓ | ✓ | | ✓ | 79.5 | 50.6 | **72.6** | 78.1 | 46.6 | **70.6** |

**Ablation study on pixel count fine-tuning and component designs.** In this part, we present an ablation study to validate the pixel count fine-tuning (*PCF*), along with the designs of the old-class features maintaining loss (*OCFM*) and the similarity-aware discriminative loss (*SAD*). In all of these experiments, our prototype replay strategy (*Replay*) and background repetition strategy (*BR*) are consistently implemented. **(1)** Table 4 displays the results of STAR with or without *PCF*. *PCF* only marginally influences the quantity of replayed or repeated samples and does not fundamentally impact their diversity. Thus, as we discussed in Sec. 2, it is an optional choice that exhibits minimal effect on the outcomes, revealing no difference in mIoU up to the first decimal place across both protocols.

**(2)** In Table 5, we compare the performance of our model equipped with *OCFM* to a model that incorporates a knowledge distillation that constrains feature changes across the entire spatial region (*KD*) [6]. It is clear from the results that our model with *OCFM* surpasses the *KD* model by 1.6 and 3.9 mIoU on the two protocols, respectively. These findings underscore the beneficial impact of the targeted nature of *OCFM*, which specifically limits alterations to old-class features. In contrast, *KD* restricts feature changes across all areas, potentially compromising the adaptability in learning new classes. **(3)** In Table 6, we examine the effects of *SAD*, which penalizes the distance between only the most similar old-new class pairs, and a discriminative loss that penalizes the average distance among all old and new class pairs (*MD*). The results highlight the superiority of *SAD* as it outperforms *MD* by 0.4 mIoU and 4.0 mIoU on the two protocols, respectively. In contrast to *MD*, *SAD* is more focused and directs differentiation effort where it is most required, thereby enhancing efficiency.

### 3.4 Qualitative results in steps

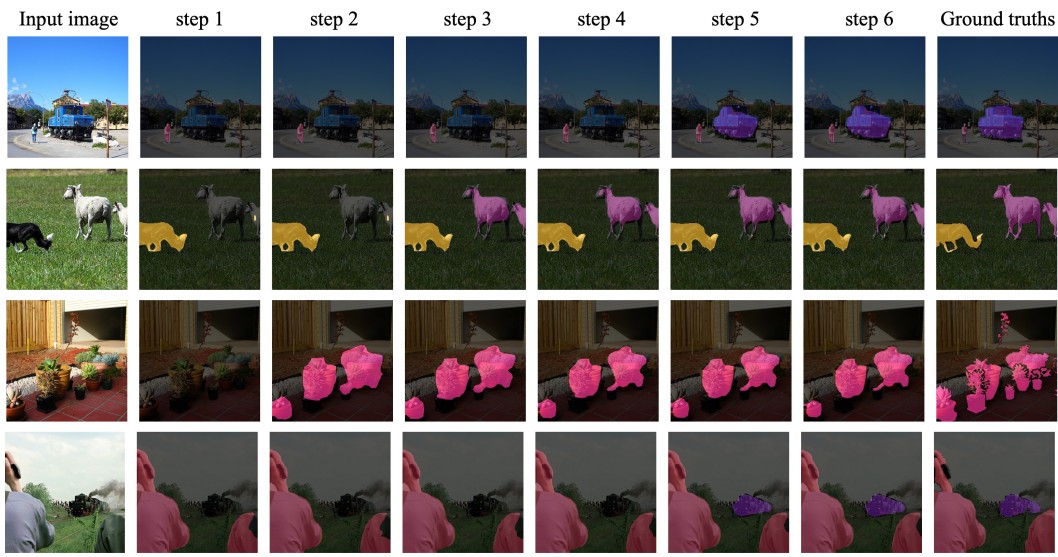

Figure 1: Qualitative results for the *overlapped* 15-1 protocol on Pascal VOC 2012. `train`, `sheep`, and `potted plant` belong to the new classes.

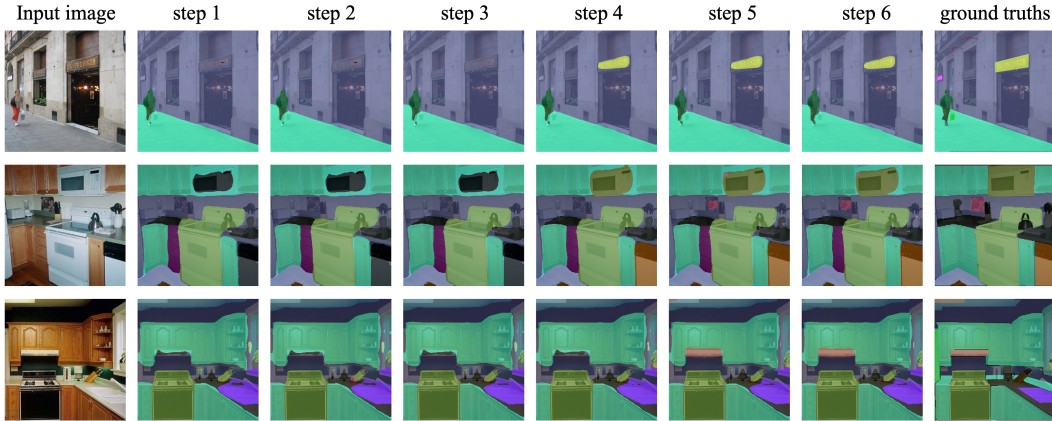

Figure 2: Qualitative results for the *overlapped* 100-10 protocol on ADE20K. `trade name`, `microwave`, `tray`, and `hood` belong to the new classes.

Fig. 1 and Fig. 2 depict the visualized predictions of STAR for the same image after each step, corresponding to the *overlapped* 15-1 protocol on the Pascal VOC 2012 dataset and the *overlapped*

100-10 protocol on the ADE20K dataset, respectively. As observed from these figures, STAR effectively learns to distinguish new classes during each incremental step while simultaneously retaining the ability to identify old classes, demonstrating no sign of forgetting.

### 3.5 Limitations and future works

Although our model demonstrates impressive performance, we acknowledge certain limitations: **(1)** We utilize Gaussian distributions to model replay samples. This empirically driven choice, however, was not subject to a detailed analysis. Despite the broad applicability of the Gaussian distribution, it might not be the most fitting selection for representing the distribution of class feature vectors. This issue invites further investigation in future research. **(2)** We preserve merely a single representation as the prototype for each old foreground class. While it is the feature center that embodies the most typical characteristics, and the added noise provides some variability, it may still lack sufficient diversity. We envision that future research could extend the number of preserved representations, each incorporating different yet typically representative characteristics, thereby enriching the diversity of replay samples. Although this could potentially increase storage consumption, it would still be substantially more efficient than storing raw images.