# OpenReview forum: "Saving 100x Storage: Prototype Replay for Reconstructing Training Sample Distribution in Class-Incremental Semantic Segmentation"
_NeurIPS.cc/2023/Conference — NeurIPS 2023 poster_

### Official Review · Reviewer_DAss · 2023-07-03

**Soundness:** 4 excellent
**Presentation:** 4 excellent
**Contribution:** 4 excellent
**Rating:** 6
**Confidence:** 5

**Summary:**

This paper proposes a memory-efficient replay-based class-incremental semantic segmentation method. Compared to the existing replay-based method, it achieves better performance while saving over 100 times the storage and can mitigate the private issue of the stored data. Furthermore, it proposes OCFM and SAD loss functions to improve plasticity and prevent catastrophic forgetting of the model. Experiments on two benchmarks (VOC 2012 and ADE 20K) verify the effectiveness and efficiency of the proposed method.

**Strengths:**

### 1. The motivation of this work is well-described in the introduction section

- I can easily agree with the problems: the discrepancies in class distributions, the biased classifier towards new classes, the memory storage efficiency issue of replay-based methods, and the private issue of the stored data.

### 2. The proposed prototype is well-designed and efficient.

- Even though the prototypes are 1-d vectors that condense the information of old classes, the Gaussian-distributed random variables can reasonably represent potential characteristics of old classes.

### 3. Thorough experiments and ablation study.

- The effectiveness of the proposed method is well-verified on the popular two benchmarks (VOC 2012 and ADE 20K).
- The ablation study shows the role of each proposed component.

**Weaknesses:**

### 1. Recommend using an Algorithm.

- Since the detailed procedure of the proposed method (Sections 2.3 and 2.4) is mostly described using manuscripts, it makes the readers to hard to understand it.
- I think it would be better to describe the proposed method using an algorithm.


### 2. Describing the multiple binary cross-entropy (mBCE) loss

- When leveraging the prototype and background repetition, what is the ground-truth? where the predictions are come from? what is the equation of the objective function?
- (This may overlap with question 1.)

**Questions:**

N/A

**Limitations:**

The authors have discussed limitations in the supplementary material.

---

> ### Author Rebuttal · Authors · 2023-08-08
>
> **Q1: Algorithm.**
> Thank you for your valuable suggestion. Below is an algorithm describing our proposed STAR. Please note that due to the constraints of this platform, we're presenting it in plain text. In the final version, we will format it using LaTeX.
>
> **Algorithm 1:**  Pseudo code of STAR
>
> **Require:** feature extractor $\Psi_\theta$, classifiers $\Phi_\omega$, number of steps $T$, number of epochs $N_{epoch}$, training sets for all steps $\\{{D^t}\\}_{t=1}^T$, foreground classes for all steps $\\{{L^t}\\} _{t=1}^T$, balance weights in the total loss $\alpha$ and $\beta$
>
> **for** $t$ in $1$ to $T$ **do**
>
> &ensp;**if** $t=1$ **then**
>
> &emsp;Random initialize $\theta$, $\omega$
>
> &ensp;**else**
>
> &emsp;Initialize $\theta$, $\\{{\omega_l}\\}_{l\in L^{1:t-1}}$ with $\theta^{t-1}$, $\\{{\omega_l^{t-1}}\\} _{l\in L^{1:t-1}}$
>
> &emsp;Random initialize $\\{{\omega_l}\\}_{l\in L^t}$
>
> &ensp;/* After initialization, $\theta$ and $\omega$ are re-denoted as $\theta^t$ and $\omega^t$ */
>
> &ensp;**end if**
>
> &ensp;**for** $n_{epoch}$ in $1$ to $N_{epoch}$ **do**
>
> &emsp;**for** batch $U$ in $D^t$ **do**
>
> &ensp;&emsp;$\Psi_{\theta^t}$ extract features, achieving $f^t$
>
> &ensp;&emsp;**if** $t=1$ **then**
>
> &emsp;&emsp;Feed $f^t$ to $\Phi_{\omega^t}$, achieving the output $p^t$
>
> &emsp;&emsp;/* mBCE loss */
>
> &emsp;&emsp;Compute $\mathcal{L}_{mbce}$ as the multiple binary cross-entropy between $p^t$ and the ground-truth
>
> &emsp;&emsp;/* parameter update */
>
> &emsp;&emsp;Compute the gradient for $\mathcal{L}_{mbce}$, then update $\theta^t$ and $\omega^t$
>
> &emsp;&ensp;**else**
>
> &emsp;&emsp;$\Psi_{\theta^{t-1}}$ extract features, producing $f^{t,t-1}$
>
> &emsp;&emsp;Feed $f^{t,t-1}$ to $\Phi_{\omega^{t-1}}$, achieving the prediction of old classes, $p^{t,t-1}$
>
> &emsp;&emsp;/* prototype replay */
>
> &emsp;&emsp;Construct Gaussian-distributed random variables $\\{{r_l}\\}_{l\in L^{1:t-1}}$ based on saved prototypes and statistics
>
> &emsp;&emsp;Sample each $r_l$ a certain number of times based on the saved occurrence count of $l$ pixels and concatenate all random samples together
>
> &emsp;&emsp;/* background repetition */
>
> &emsp;&emsp;Identify background regions by excluding pixels currently labeled as foreground class and old-class pixels estimated in $p^{t,t-1}$
>
> &emsp;&emsp;Duplicate background pixels a certain number of times based on the saved occurrence count of background pixels and concatenate all duplications together
>
> &emsp;&emsp;/* mBCE loss */
>
> &emsp;&emsp;Concatenate aforementioned random samples and background duplications with $f^t$, feed them to $\Phi_{\omega^t}$, achieving the output $p^t$
>
> &emsp;&emsp;Compute $\mathcal{L}_{mbce}$ as the multiple binary cross-entropy between $p^t$ and corresponding ground-truths // refer to Q2 response for details
>
> &emsp;&emsp;/* old-class features maintaining loss */
>
> &emsp;&emsp;Identify old-class regions with the help of $p^{t,t-1}$
>
> &emsp;&emsp;Compute $\mathcal{L}_{ocfm}$ as the mean square error between $f^t$ and $f^{t,t-1}$ in old-class regions
>
> &emsp;&emsp;/* similarity-aware discriminative loss */
>
> &emsp;&emsp;Compute $\mathcal{L}_{sad}$ as the mean of the minimum distance from the feature center of each current foreground class (computed within the current batch) to all saved old-class prototypes
>
> &emsp;&emsp;/* parameter update */
>
> &emsp;&emsp;Compute the gradient for $\mathcal{L}_{mbce}+\alpha\mathcal{L} _{ocfm}+\beta\mathcal{L} _{sad}$, then update $\theta^t$ and $\omega^t$
>
> &emsp;&ensp;**end if**
>
> &emsp;**end for**
>
> &ensp;**end for**
>
> &ensp;Compute and save the occurrence counts for all current foreground class pixels and background pixels
>
> &ensp;Compute and save the prototypes and statistics of pixel-level features for all current foreground classes
>
> **end for**
>
> return $\Psi_{\theta^T}$ and $\Phi_{\omega^T}$
>
> **Q2: mBCE loss.**
>
> We use a standard mBCE loss computed as: $\mathcal{L} _{mbce}=-\frac{1}{H\cdot W+N _{rp}+N _{rb}}\sum _{j=1}^{H\cdot W+N _{rp}+N _{rb}}\sum _{l\in L^t}{\gamma\cdot y _j^{t,l}\cdot\log{p _j^{t,l}}+(1-y _j^{t,l})\cdot\log{\left(1-p _j^{t,l}\right)}}$. Here, $H\cdot W$ indicates the spatial size of the input image, $N _{rp}$ is the number of replayed prototype samples, $N _{rb}$ is the number of repeated background pixels, $j$ is the index, $L^t$ is the set of current foreground classes, $t$ is the step index, $\gamma$ is the positive weight, $y _j^{t,l}$ is the ground-truth of the $j$-th pixel or replayed sample ($0$ means it does not belong to class $l$, $1$ means it does), and $p_j^{t,l}$ is the network's output. In the first step, $N _{rp}=N _{rb}=0$, and $p$ and $y$ only corresponds to the pixels in the input image. In later steps, $p$ and $y$ are the concatenations of the prediction/ground-truth for the input image, replayed prototypes, and repeated background pixels.
>
> Q2-1: When leveraging prototype replay and background repetition, the ground-truth for a replayed prototype sample is the one-hot encoding of the prototype's class label. Meanwhile, the ground-truth of each repeated background pixel is a all-zero vector, as the background does not belong to any class in $L_t$.
>
> Q2-2: During inference, the input image first goes through the feature extractor to produce features. Then, the classifiers output logits for each pixel, which are then passed through a sigmoid function to determine the probability of belonging to every foreground class. If none of the foreground class probabilities exceed $0.5$ for a pixel, the prediction of this pixel is "background". Otherwise, the prediction is the class with the highest probability.
>
> Q2-3: The overall objective function of the network is
> $\mathcal{L}=\mathcal{L} _{mbce}+\alpha\mathcal{L} _{ocfm}+\beta\mathcal{L} _{sad}$, where $\mathcal{L} _{mbce}$ is the mBCE loss as mentioned before. $\mathcal{L} _{ofcm}$ and $\mathcal{L} _{sad}$ are described in formulas (10) and (12) in the manuscript.

---

> > ### Comment · Reviewer_DAss · 2023-08-21
> > **Dear authors**
> >
> > Dear authors,
> >
> > Thank you for the thoughtful rebuttal comments. Most of my concerns are well-addressed, and I conclude that this paper is ready for publication on NeurIPS. I will keep my positive rating.

---

> > > ### Author Response · Authors · 2023-08-21
> > > **Thank you**
> > >
> > > Thank you for your positive feedback and the highly beneficial suggestions you provided, which have greatly benefited our work.
> > >
> > > Best wishes,
> > > Authors

---

### Official Review · Reviewer_2RZD · 2023-07-03

**Soundness:** 3 good
**Presentation:** 3 good
**Contribution:** 3 good
**Rating:** 8
**Confidence:** 5

**Summary:**

A new method called STAR is introduced in this paper to address the CISS task. The fundamental idea is to remove the distribution disparity between single-step training samples and the entire dataset, which consequently eliminate the network's bias. Based on this idea, the authors designed two strategies, namely the prototype replay strategy and background repetition strategy. Both strategies help to rectify the distribution of single-step training samples while keeping the storage cost minimal. Additionally, the OCFM loss is proposed to maintain features of previous foreground classes, and the SAD loss is devised to help distinguish similar categories. Experimental results demonstrate that the network achieves state-of-the-art performance, and each module contributes to its effectiveness.

**Strengths:**

From my perspective within this field, the idea presented in this paper is innovative.
The proposed approach is commendable as it effectively addresses the problem of distribution differences while keeping the storage cost low, as demonstrated by the experimental results.
Moreover, apart from CISS, the motivation and core idea have the potential to be extended to other continual learning tasks.
The overall writing of this paper is also clear, making it relatively easy to comprehend.

**Weaknesses:**

There are some grammar errors and unclear explanations in the article that may hinder readers' comprehension.
1. Firstly, in Section 1, the sentence "under the overlapped 19-1 protocol, ..." is mentioned, but it does not provide a clear explanation of which dataset this protocol refers to.
2. Secondly, in Section 2.3, the word “contain” in the sentence "Assuming each epoch contain e iterations" should be corrected to "contains" since it is in the third person singular form.

**Questions:**

Please address the issues mentioned in the weaknesses section and make modifications accordingly.

**Limitations:**

The author appropriately discusses the limitations of the proposed method in the supplementary material.

---

> ### Author Rebuttal · Authors · 2023-08-08
>
> **Q1: Unclear protocol description.**
>
> The "overlapped 19-1" protocol is a CISS protocol on the Pascal VOC 2012 dataset, where the first training step includes 19 foreground classes, while the second training step covers only one foreground class (tv/monitor). We will provide a clearer explanation of the "overlapped 19-1" protocol in the revised manuscript.
>
> **Q2: Grammatical errors.**
>
> Thank you for careful reading. We will correct this sentence to “Assuming each epoch contains $e$ iterations, …”. Moreover, we will proofread our manuscript again to avoid similar grammar errors.

---

> > ### Comment · Reviewer_2RZD · 2023-08-20
> > **Response to author rebuttal**
> >
> > I've read the author's rebuttal to me and other reviewers. All concerns have been addressed, and I didn't find any new issues. In general, the paper offers an innovative method to tackle CISS and shows good results. The overall writing is clear, making it easy to grasp the proposed approach. As a result, this paper is of high-quality, and I believe it fits the standards of NeurIPS.

---

> > > ### Author Response · Authors · 2023-08-21
> > > **Thank you**
> > >
> > > We appreciate your full recognition of our contributions, and we're truly grateful for the high praise you've given our work.
> > >
> > > Best wishes,
> > > Authors

---

### Official Review · Reviewer_gzuL · 2023-07-04

**Soundness:** 3 good
**Presentation:** 3 good
**Contribution:** 3 good
**Rating:** 7
**Confidence:** 4

**Summary:**

This paper proposes a class-incremental semantic segmentation (CISS) method, STAR, which incorporates four contributions.

[1] Prototype preservation and replay: Aligning the proportion of old-class pixels in the single-step training set with the proportion in the complete dataset by replaying old-class prototypes.

[2] Background repetition: Aligning the proportion of background in the single-step training set with the proportion in the complete dataset by repeat background pixels.

[3] Old-class features maintaining loss: Maintaining old-class features to avoid the saved prototypes from ineffective.

[4] Similarity-aware discriminative loss: Enhancing the feature distinction between similar old-new class pairs.
By combining these four components, STAR achieves good performance.


**Strengths:**

+ The primary motivation of eliminating the distribution difference between single-step training set and the complete dataset is reasonable. The strategies of prototype replay and background repetition are proper to solve this problem.
+ The proposed OCFM loss can mitigate catastrophic forgetting as well as keep the saved old-class prototypes effective. The proposed SAD loss can help the classifiers to discriminate similar old-new class pairs.
+ Compared with previous CISS methods, the performance is top-tier, and the storage requirement is negligible. The experiments are complete and capable of proving the effectiveness of the proposed method.


**Weaknesses:**

[1] In some equations, the elementwise multiplication is strange. For example, in equation (3), it seems that $\mathbb{1}\{\tilde{y}_{i,j}^{t-\tau}=l\}$ is a single element, not a vector, so how can it be element-wise multiplied with other terms? Similar situations also appear in other equations.

[2] In line 120, you mention "the complete dataset accumulated up to this step". What does it mean? Is it the same as "the union of all training sets up to this step" in line 118? If they are the same, it is strange to say the distributions on them are “close”.

[3] The pixel count fine-tuning part and limitation are introduced in the supplementary due to the page limitation, but authors should at least give a brief introduction in it to outline your basic principle of it.


**Questions:**

See the weakness

**Limitations:**

This paper addresses the issue of biased classifiers observed in previous methods, and it discusses the limitations of the proposed method as well as areas for improvement properly.

---

> ### Author Rebuttal · Authors · 2023-08-08
>
> **Q1: Unclear equations.**
>
> Thank you for comments. Indeed, $\mathbb{1}\\{\tilde{y}_{i,j}^{t-\tau}=l\\}$ is a single element, not a vector, so it can only multiply, other than element-wise multiply other terms. We will amend this equation to  $\mu_l = \frac{\sum _{i=1}^{N^{t-\tau}} \sum _{j=1}^{h\times w} (\hat{f} _{i,j}^{t-\tau} \mathbb{1} \\{\tilde{y} _{i,j}^{t-\tau}=l\\} )}
> {\left\lVert
> \sum _{i=1}^{N^{t-\tau}} \sum _{j=1}^{h\times w} ( \hat{f} _{i,j}^{t-\tau} \mathbb{1} \\{\tilde{y} _{i,j}^{t-\tau}=l\\} )
> \right\rVert_2}
> \in \mathbb{R}^c$. Since the similar situation also appears in other equations, we will also modify them accordingly.
>
>
> **Q2: Unclear descriptions.**
>
> The former encompasses all training samples encountered up to the current step, along with corresponding test samples. The latter, however, exclusively represents the collection of training samples encountered thus far. Essentially, the latter is a subset of the former. Since the training set and test set are usually randomly partitioned, their distributions should be close.
>
> **Q3: Briefly describe the pixel count fine-tuning and limitations in the main body.**
>
> Thank you for your suggestion. We will incorporate brief introductions about our pixel count fine-tuning strategy in Sections 2.3 and 2.4 to enhance the readability.

---

> > ### Comment · Reviewer_gzuL · 2023-08-20
> > **Response to Authors**
> >
> > Thank you for addressing my questions and correcting the typos. Overall, this work is good with a sound motivation, proper solutions, and superior performance. Hence, I will maintain my positive rating.

---

> > > ### Author Response · Authors · 2023-08-21
> > > **Thank you**
> > >
> > > Thank you for your positive assessment and your meticulous review. Your feedback has been very helpful to us.
> > >
> > > Best wishes,
> > > Authors

---

### Official Review · Reviewer_b1on · 2023-07-05

**Soundness:** 3 good
**Presentation:** 3 good
**Contribution:** 2 fair
**Rating:** 5
**Confidence:** 5

**Summary:**

The topic of this paper is about Class-Incremental Semantic Segmentation (CISS). The authors focus on another challenge of CISS: the discrepancies in class distributions between single-step training sets and the complete dataset. They propose a STAR (Storage sAving Replay) framework including a prototype replay strategy, background repetition strategy, old-class feature maintaining loss and similarity-aware discriminative loss.

**Strengths:**

+ Exploring the biased distribution problem in CISS is interesting.
+ The proposed method achieves a large performance gain on several datasets.


**Weaknesses:**

1. The practicability of the proposed method seems to be limited. The authors claim that the distribution of the single-step training set should be similar as that of the complete dataset. However, the distribution of the complete dataset may be not available practical incremental learning scenarios (we can not get the information of newly arrived data).

2. The key novelty of the prototype replay strategy is not clear. The authors claim that the proposed prototype replay strategy can greatly save the memory cost. In the class-incremental learning community, several works focus on the memory efficiency of rehearsal samples (e.g. [1]-[4]).

[1] Memory-Efficient Incremental Learning Through Feature Adaptation, ECCV2020

[2] Semantic Drift Compensation for Class-Incremental Learning, CVPR2020

[3] Memory efficient class-incremental learning for image classification, TNNLS2021

[4] A Model or 603 Exemplars: Towards Memory-Efficient Class-Incremental Learning, ICLR2023

3. Concerns about the effectiveness of the saved prototypes. As described in SDC[2], the information of the stored old-class prototype may be imprecise after the model is updated on the new data.

4. Some related works should be included in the paper. The new-class-biased distribution problem has been discussed in some CISS or image CIL works (e.g. [5]-[6]). These works have the similar motivation as this paper.

[5] Large Scale Incremental Learning, CVPR2019

[6] RBC: Rectifying the Biased Context in Continual Semantic Segmentation, ECCV2022

**Questions:**

Q1: How to get the distribution of the complete dataset in practical CISS scenario?

Q2: What is the key difference between the proposed prototype replay strategy and other classic memory-efficient CIL methods?

Q3: For the STAR framework, how to alleviate the drift of the stored old-class prototype?

Q4: The key biased distribution problem in CISS and the corresponding works should be sumarized in a related work section.


**Limitations:**

The authors have addressed the limitations.

---

> ### Author Rebuttal · Authors · 2023-08-08
>
> **Q1: The practicability of our work.**
>
> As you pointed out, we cannot know the information of newly arrived data in practical scenarios. Therefore, our experiments simulate this real-world situation, without considering the distribution of future data. Actually, when we mention the "complete dataset", it involves all the training samples encountered up to the present step and corresponding test samples, excluding future data. As stated in Section 2.3 of the manuscript, our goal is to align the class distribution of current single-step training samples with that of "all training sets up to this step". This distribution should resemble that of the "complete dataset accumulated up to this step" since training and test set distributions are typically comparable. With this goal achieved, the classifiers' tendency can closely resemble the outcomes in joint training on all seen training samples, which is the theoretical upper limit of continual learning.
>
> We admit that our description of the term "complete dataset" may not be clear, possibly leading to misunderstandings. Thus, we will offer a more explicit clarification in the revised version.
>
> **Q2: The key novelty of our prototype replay strategy.**
>
> As you mentioned, some CIL works also focused on enhancing memory efficiency, including [1, 2, 3, 4] that you referred to. First, as for [4], the focus is memory efficiency of model parameters, which is entirely different from our study and [1, 2, 3]. For CISS, model-based methods are rarely used, so most models have the same parameter count, rendering the concern explored in [4] does not really apply. For the remaining works, [1, 2] and ours utilize compact prototypes to retain old class information. Both [1, 2] concentrated on the issue of prototype ineffectiveness caused by feature space changes during incremental training and proposed different methods to adapt previously stored prototypes to the current feature space. In contrast, our approach exploits the specific characteristics of CISS (refer to Q3 response for details) to maintain the old-class feature space, providing a different perspective to address this issue. For [1, 2], insufficient prototype adaptation could still result in prototypes losing part of effectiveness, a problem does not present in our method. On the other hand, [3] used the low-fidelity exemplars rather than prototypes. Their primary concern is narrowing the domain gap between low-fidelity exemplars and raw images, which is quite different from our work.
>
> Additionally, each prototype in [1] or each low-fidelity exemplar in [3] corresponds to an individual image, leading to suboptimal memory efficiency since multiple prototypes or exemplars must be stored per class for robust training. Conversely, our approach and [2] implement one prototype per class, resulting in lower memory cost. Nonetheless, our method differs from [2] in that we store both prototypes and the statistics of pixel-level features for each class. This equips us with more comprehensive cues for old classes in subsequent steps, resulting in robust training outcomes. Note that, despite storing the statistics, our total memory cost remains comparable to [2] as our prototypes are more compact. Moreover, our attention to class distribution and the strategy of replaying prototypes based on it distinguish our approach. This strategy enables us to effectively correct the network's bias towards new classes, an issue overlooked in [1]-[3].
>
> Lastly, another fundamental difference lies in the particular tasks we tackle compared to [1, 2, 3, 4]. While all fall under the umbrella of CIL, [1, 2, 3, 4] addressed classification but our focus is segmentation (CISS). Given the paucity of study on memory efficiency within the CISS domain, we believe our contribution holds innovative value.
>
> In summary, our method is different from prior memory-efficient CIL methods in four aspects: (1) we save a single compact prototype per class, achieving better memory efficiency. (2) Our strategy incorporates class distribution during replay, effectively rectifying classifier bias. (3) We store feature statistics beyond prototypes, offering more comprehensive cues of past classes. (4) The specific task we address is different. We will clarify the difference from these methods in the final version.
>
> **Q3: The alleviation of drift.**
>
> The concept of "drift" presented in SDC [2] refers to the shift of the feature space during incremental learning, which impedes the effectiveness of prototypes calculated in past feature spaces. In our method, solving this problem is also crucial. Thus, we devise a solution tailored to address it in the CISS task, i.e., the Old-Class Features Maintaining (OCFM) loss. Compared to the classification task of CIL, one difference of CISS is the potential presence of old-class pixels in the current training images (although labeled as background).  Our solution takes advantage of this characteristic, maintaining the features in old-class regions unchanged while allowing other features to update. Specifically, we first use the model from the previous step to locate old-class pixels. Next, the features extracted by the current model at these old-class pixels are constrained to match those extracted by the previous model, thus maintaining the feature space of old classes and avoiding drift. Our ablation study results in Section 3.3 show that the OCFM loss can effectively alleviate drift, ensuring the efficacy of prototypes.
>
> **Q4: Missing some related works.**
>
> We appreciate your suggestion. Due to the page limit of the initial submission, we briefly addressed related works in the first and third paragraphs of the "Introduction" section. In the final version, with one more page permitted, we will add a separate "Related Works" section to offer a more comprehensive summary of related works, including those also focus on the biased distribution problem you mentioned.

---

> > ### Comment · Reviewer_b1on · 2023-08-22
> > **Thanks for the response**
> >
> > Thank you for the response. The authors have carefully addressed all of my concerns, I have no other issues and decide to keep my initial score.

---

> ### Author Response · Authors · 2023-08-21
> **Thank you**
>
> We appreciate your thoughtful and highly constructive feedback, which has allowed us to enhance the quality of our manuscript.
>
> Best wishes,
> Authors

---

### Official Review · Reviewer_YBrR · 2023-07-07

**Soundness:** 4 excellent
**Presentation:** 3 good
**Contribution:** 2 fair
**Rating:** 6
**Confidence:** 5

**Summary:**

The paper utilizes prototypes to store information which saves over 100 times the storage while achieving better performance. It also, introduces old-class features maintaining loss and a similarity-aware discriminative loss.

**Strengths:**

1. The performance of this paper is exciting as it achieves SOTA performance while saving tons of memory compared to existing replay based methods.

2. The code of the paper is provided and detailed implementation details are given, which makes the paper easy to replicate.

3. The paper is overall clearly written and easy to follow.

**Weaknesses:**

1. The use of prototypes to store past information is widely used in CSS including the paper SDR. It would be great if the authors can make a clear distinguish of the novelty of this paper and previous paper using prototypes.

2. The issue of discrepancies in class distributions between single-step training sets and the complete dataset (line 42) is in fact already proposed in the paper RBC: Rectifying the Biased Context in Continual Semantic Segmentation (ECCV 2022), the authors can consider citing it.

**Questions:**

NA

---

> ### Author Rebuttal · Authors · 2023-08-08
>
> **Q1: The novelty of prototype usage.**
>
> As you mentioned, some prior CSS works have employed prototypes to save past information, with a representative being SDR that you referred to. However, there is a clear distinction in the specific usage of prototypes between SDR and our STAR. In SDR, the primary functions of prototypes are to mitigate catastrophic forgetting and enhance the consistency of features within the same class. Specifically, in each batch, features of each old class are used to generate a "current prototype". Then, a specialized loss function is employed to constrain this "current prototype" closely match the saved prototype of the corresponding class, aiming to stabilize old-class features and alleviate catastrophic forgetting. Simultaneously, pixel-level features of each foreground class are pulled towards their corresponding prototype, promoting feature cohesion within the same class.
>
> Contrary to SDR, our STAR primarily employs prototypes for replay, aiming to rectify the classifiers' bias towards new classes. Specifically, in each training step, we save the prototypes, pixel-level feature statistics, and pixel occurrence counts for each class. For subsequent steps, we establish a random variable for every prior class using its prototype and feature statistics, which mirrors potential feature representations for that class. We then sample from this variable based on the pixel occurrence count of its class and input these random samples to the classifiers. This procedure aligns the class distribution of the single step training samples with the cumulative training set up to the current point, mitigating classifier bias. In summary, our strategy for prototype utilization diverges from SDR's both in purpose (combatting forgetting vs. bias reduction) and approach (feature constraints vs. replay). Moreover, our modeling of prototypes as statistics-driven random variables, along with our alignment of class distributions in single-step training samples, fundamentally distinguishes our approach from previous studies. From the results of our comparative experiments and ablation study, it can be seen that our prototype replay strategy is effective, contributing to the satisfactory overall performance of our STAR.
>
> **Q2: Citing the RBC paper.**
>
> Thank you for providing the reference. Indeed, RBC also tackles the issue of discrepancies in class distributions between single-step training sets and the complete dataset. We will cite and briefly discuss this paper in the final version.
>
> The proposed solution of RBC is twofold. First, during incremental training, it employs pairs of input images, one normal and the other with new-class pixels erased. Second, it enhances the weight of old-class pixels in the loss function. The first strategy is equivalent to reducing half of the new-class pixels, while the second strategy enhance the weights of old classes in the loss function based on the ratio of new-to-old classes in the current image. These strategies do help mitigate the bias of classifiers towards new classes, producing improved results. However, we believe that there is still room for further refinement. This is because RBC does not take into account the actual class proportions within the complete dataset when rectifying the bias. Thus, while it reduces the bias towards new classes, it does not guarantee that the classifiers' tendency is adjusted to an appropriate level. From this perspective, our method, which replays old-class prototypes based on the recorded class proportion in the training samples accumulated up to the present, holds an advantage, because it can bring the classifiers' tendency closer to the level achieved in joint training on these samples. Meanwhile, the introduction of statistics-driven noise into replayed samples ensures a comprehensive incorporation of past classes.

---

> ### Author Response · Authors · 2023-08-21
> **Thank you**
>
> Thank you for reviewing our manuscript and for your valuable comments, which have been greatly beneficial to us.
>
> Best wishes,
> Authors

---

### Decision · Program_Chairs · 2023-09-21

**Decision:**

Accept (poster)

**Comment:**

This paper received all five positive scores. In the initial review procedure, some common concerns are about the motivation novelty and comparison with existing works. After rebuttal, reviewers agree that these concerns are well addressed. After checking the submission and all reviews, the AC agrees that this paper meets the requirement of the NeurIPS, and recommends acceptance.